# Cyano-B12 or Whey Powder with Endogenous Hydroxo-B12 for Supplementation in B12 Deficient Lactovegetarians

**DOI:** 10.3390/nu11102382

**Published:** 2019-10-06

**Authors:** Sadanand Naik, Namita Mahalle, Eva Greibe, Marie S. Ostenfeld, Christian W. Heegaard, Ebba Nexo, Sergey N. Fedosov

**Affiliations:** 1Department of Pathology, Deenanath Mangeshkar Hospital and Research Center, Pune 411004, India; pnmahalle@gmail.com; 2Department of Clinical Biochemistry and Institute of Clinical Medicine, Aarhus University Hospital, DK-8200 Aarhus N, Denmark; evagreibe@gmail.com (E.G.); ebbanexo@rm.dk (E.N.); 3Arla Foods Ingredients Group P/S, DK-8260 Viby, Denmark; marie.stampe.ostenfeld@arlafoods.com; 4Department of Molecular Biology and Genetics, Aarhus University, DK-8000 Aarhus C, Denmark; cwh@mbg.au.dk

**Keywords:** vitamin B12, cyano-B12, hydroxo-B12, B12 supplementation, B12 deficiency, milk, whey powder, lactovegetarians

## Abstract

Lactovegetarians (*n* = 35) with low vitamin B12 (B12) status were intervened for eight weeks capsules containing cyano-B12 (CN-B12), (2 × 2.8 µg/day), or equivalent doses of endogenous B12 (mainly hydroxo-B12 (HO-B12)) in whey powder. Blood samples were examined at baseline, every second week during the intervention, and two weeks post-intervention. The groups did not differ at baseline in [global median (min/max)] plasma B12 [112(61/185)] pmol/L, holotranscobalamin [20(4/99)] pmol/L, folate [13(11/16)], the metabolites total homocysteine [18(9/52)] µmol/L and methylmalonic acid [0.90(0.28/2.5)] µmol/L, and the combined indicator of B12 status (4cB12) [−1.7(−3.0/−0.33)]. Both supplements caused significant effects, though none of the biomarkers returned to normal values. Total plasma B12 showed a higher increase in the capsule group compared to the whey powder group (*p* = 0.02). However, the increase of plasma holotranscobalamin (*p* = 0.06) and the lowering of the metabolites (*p* > 0.07) were alike in both groups. Thereby, the high total plasma B12 in the capsule group was not mirrored in enhanced B12 metabolism, possibly because the B12 surplus was mainly accumulated on an “inert” carrier haptocorrin, considered to be of marginal importance for tissue delivery of B12. In conclusion, we demonstrate that administration of whey powder (HO-B12) or capsules (CN-B12) equivalent to 5.6 µg of B12 daily for eight weeks similarly improves B12 status but does not normalize it. We document that the results for plasma B12 should be interpreted with caution following administration of CN-B12, since the change is disproportionately high compared to the responses of complementary biomarkers.

## 1. Introduction

Vitamin B12 (B12, cobalamin) is one of the greatest nutritional concerns for vegetarians as vitamin B12 is primarily present in foods of animal origin [1]. B12 deficiency is identified by a decrease in total plasma B12, as well as by B12 bound to its transport protein transcobalamin (holotranscobalamin, holoTC) and increase in the metabolites–plasma homocysteine (Hcy) and methylmalonic acid (MMA) [2,3]. Increased concentrations of MMA and Hcy are due to decreased activity of methylmalonyl-CoA mutase and methoionine synthase, respectively [1,2,3,4,5]. While severe deficiency can cause megaloblastic anemia and permanent neurological damage, early physiological manifestations are generally subtle [1,6].

Two main etiologic factors play a role in developing vitamin B12 deficiency: Inadequate dietary intake and/or vitamin B12 malabsorption. Asymptomatic Indian lactovegetarians, who make up more than half of the Indian population, have distinctly lower vitamin B12 concentrations than non-vegetarians across geographic regions of India [7,8]. These lactovegetarians were also found to have increased plasma levels of Hcy and MMA, as well as lowered holoTC, all indicating vitamin B12 deficiency [7,8]. The options of increasing the overall vitamin B12 status of a deficient population include vitamin supplementations, fortification or targeted dietary recommendations.

Epidemiological and observational studies show that B12 status correlates with the consumption of animal products [5,6,7,8,9,10]. Among those, endogenous B12 in dairy products and especially milk appears to be highly bioavailable [11]. In accordance, intervention studies have revealed that the B12 status of vegetarians is positively associated with their intake of milk [7,8] and that both cow and buffalo milk [12,13] match supplements with B12 capsules in improving biomarkers of B12 deficiency [8]. The native form of B12 in milk is mainly hydroxo-B12 (HO-B12) [8], whereas the synthetic variant cyano-B12 (CN-B12) is widely used in supplements and fortified foods. Employing an animal model, we have previously demonstrated that, though the two forms of B12 are absorbed alike, they distribute differently in the body [14,15]. CN-B12 accumulated mainly in the kidneys, whereas HO-B12 predominantly targeted the liver. In both human and animal studies, CN-B12 supplementation caused the highest increase in plasma B12, yet more biologically active B12 coenzymes were retrieved from the tissues of the HO-B12 supplemented animals [16]. These findings question whether the two vitamin forms are of equal value for the improvement of B12 status.

A recent eight weeks intervention study in elderly Australians with subclinical deficiency of B12 revealed that cow’s milk whey powder, unlike devoid of B12 soy protein isolate, increased plasma holoTC and attenuated the buildup of MMA and Hcy [17]. Studies, carried out so far, leave us with a question, how to normalize and maintain B12 status in lactovegetarians with a borderline B12 deficiency. The purpose of this investigation was to directly compare the efficacy of treatment with whey powder and CN-B12 in capsules (both equivalent to 5.6 µg of B12).

## 2. Materials and Methods

### 2.1. Study Design and Supplementation Products

Indian lactovegetarians were supplemented for eight weeks with two daily doses of 2.8 µg CN-B12 in capsules vs. equivalent doses of endogenous HO-B12 in whey powder. Non-fasting blood samples were drawn at baseline, each second week throughout the intervention, and again after two weeks follow-up without supplementations (see Figure 1). CN-B12 capsules were prepared at Aarhus University Hospital as previously described [18] and the whey powder (Lacprodan^®^ DI-8290) was supplied by Arla Foods Ingredients Group P/S, 8260 Viby, Denmark (for the nutritional compositions, see Appendix A
Appendix A). The whey powder (2 × 50 g) was suspended in 200 mL of water with two teaspoons of Hershey chocolate syrup and consumed in the morning and the evening. The capsules were consumed with the same frequency and suspended in the same way as the powder. Using WhatsApp, each participant was contacted in the morning and in the evening to ascertain the participants were consuming as per schedule and did not miss any dose of capsule/whey powder intake.

### 2.2. Participants

Participants (*n* = 35 after exclusion of dropouts, 12 males and 23 females, the age of 18–50 years) were recruited from the Pune area of India, and the study was carried out at Deenanath Mangeshkar Hospital and Research Center, Pune, from September 2017 to December 2017. The inclusion criterion was lactovegetarian diet. Exclusion criteria were intake of vitamin pills (containing > 1 µg B12) within the last two weeks of the study, use of drugs (methotrexate, antacids, and metformin) known to influence B12 absorption/biomarkers, and presence of known chronic systemic diseases. The number of participants (initial *n* = 40) was adjusted for expected dropouts due to the nature of the eight weeks treatment. The participants were divided into two groups, (i.e., capsule group and whey powder group), equally distributed with respect to age, and were allowed to continue on their regular lactovegetarian diet throughout the study. The participants received no compensation.

All individuals were examined at baseline, stipulating folate and B12 deficiency as serum/plasma concentrations <4.54 nmol/L and <150 pmol/L, respectively [7]. Anemia was defined as a hemoglobin concentration <7.5 nmol/L (120 g/L) in females and <8.0 nmol/L (130 g/L) in males. Macrocytosis was stipulated as MCV > 100 fl [19].

The investigation was performed within the confines of the Helsinki Declaration II, and the study was approved by the Institutional Ethics Committee of Deenanath Mangeshkar Hospital and Research Center (Ethical approval 27/09/2017, IH_2017_Aug_SN_230). All individuals gave their informed consent before inclusion in the study. The study is registered with the Clinical Trial Registry of India bearing CTRI/2018/05/03616 (ref 2/5/2018).

The overall design of the study is depicted in Figure 1.

### 2.3. Blood Sampling and Biochemical Methods

At baseline and every second week throughout the study, non-fasting blood samples were drawn into EDTA and plain vacutainers and kept in an ice-cold box. Hemoglobin and red blood cell mean corpuscular volume was determined on the XN 3000 hematology analyzer (Sysmex) [7]. Blood samples were centrifuged (10 min at 2300× *g*) within two hours after withdrawal, and serum/plasma was stored at −56 °C until analysis/shipment of samples. Plasma B12 (cobalamin, B12), holotranscobalamin (holoTC), total homocysteine (Hcy), creatinine and serum folate concentrations were measured on the Architect ci 4100 (Abbott) every two weeks in batches, each batch having internal quality control samples [7]. Plasma aliquots were shipped to Aarhus University Hospital, Aarhus, Denmark, on dry ice for analysis of methylmalonic acid (MMA), measured in one run for all participants. MMA was quantified by liquid chromatography–tandem mass spectrometry on the AB SCIEX Triple Quad 5500 System (AB SCIEX) [8]. Intra-batch coefficients of variation (in percent) for vitamin B12, holoTC, Hcy, folate and MMA were 4.8%, 6.2%, 4.7%, 4.8%, 3.2%, and 5.4% and inter-batch coefficients were 5.3%, 6.9%, 4.9%, 3.4%, and 6.2%, respectively.

In order to explore the different forms of B12 present in plasma, we prepared pools of 100 µL plasma drawn at week 8 and week 10 from five individuals in each of the two intervention groups (capsule and whey powder). Employing the previously described methods, transcobalamin, and haptocorrin bound pools of B12 were isolated by precipitation on magnetic beads coated with anti-transcobalamin IgG and anti-haptocorrin IgG, respectively. The B12 forms were liberated by enzymatic digestion, separated by HPLC and subsequently quantified by ELISA-based B12 assay [20,21].

### 2.4. Combined Analysis of Markers

The four-component combined indicator of B12 status 4cB12 was calculated from the measurements of plasma B12, holoTC, MMA, and Hcy using the formula presented by Fedosov et al. [22], which takes into account the effect of folate on Hcy. The two-component indexes 2cB12_B12,holoTC_ or 2cB12_metabolic_ included the measurements of either total plasma B12 and holoTC or the metabolites Hcy and MMA, respectively [22].

### 2.5. Statistical Procedures and Software

Examination of the baseline identity between the two supplementation groups (for each given marker) was conducted by non-parametric Wilcoxon (Mann–Whitney) test for unpaired data. The outcome of the treatment groups, (baseline vs. weeks four and eight), was assessed by Wilcoxon (Mann–Whitney) signed rank test for paired groups. The shift of scaled responses (Section 2.6) from the baseline at each time point was assessed by the parametric paired *t*-test (to maintain consistency with the parametric method of least squares also used in these charts). The pairwise identity of the scaled responses (capsules vs. whey powder) at each particular time point was examined by *t*-test. Comparison of the overall responses was done via parametric analysis of the approximation curves, see Section 2.6. All computations were performed using free software KyPlot 5 (www.kyenslab.com, Tokyo, Japan).

### 2.6. Correction for Baselines, Data Approximation, and Fitting Statistics

All responses were related to their baselines. Time-dependent shifts (Δ*y*) in total plasma B12 and holoTC (*y*) from their respective baselines (*y*_0_) were calculated for each patient as Δ*y* = *y* − *y*_0_. This scaling was expected to suppress the excessive dispersion caused by different starting points. Changes in MMA and Hcy over time were presented as ratios (*y*/*y*_0_) between the concentration at a given time point (*y*) and the concentration at the baseline (*y*_0_). Such normalization partially compensated larger (smaller) responses at higher (lower) initial metabolites, usually observed at identical changes in the B12 status, see also [8].

The obtained scaled responses were approximated by either an exponent, Equation (1) or a variant of Burr function, Equation (2), capable to reproduce the shape of the experimental data but claiming no physiological meaning.
(1)$$y\;=\;P_{1}\;+\;P_{2}\left(1\;-\;e^{-P_{3}\cdot x}\right)$$
(2)$$y\;=\;P_{1\;}\;+\;2\cdot P_{2}\cdot \frac{x/P_{3}}{{\left(1\;+\;{\left(x/P_{3}\right)}^{2}\right)}^{2}}$$

The approximating functions were used to generally assess the shift from baseline in course of capsule or whey powder treatment, as well as to compare the overall identity of the two treatments. In the first case, parameters of best approximation for each fitting function were compared to their reference values at baseline (0 or 1 depending on the context). The general shift from baseline was assessed by *t*-test, where the probability (*p*) of a “zero function” (with all parameters equal to their reference values, 0 or 1) was calculated as the product of probabilities (*p* = *p*_1_·*p*_2_·*p*_3_) for each “zero parameter” (*p*_i_). The difference in responses between the two B12 supplements was assessed by pairwise comparison of the respective parameters of best approximation (capsules vs. whey powder), whereupon probabilities of identical pairs were multiplied. The procedure was explained in detail elsewhere [8].

## 3. Results

### 3.1. Characteristics of Participants

The daily diet of the participants included dairy products but not meat/fish. The cobalamin content of the dairy products consumed was 0.6 µg/day. Thirty-five lactovegetarians (12 males and 23 females) from randomly assigned groups were subjected to an eight-week intervention and accomplished the trial (Figure 1).

The participants had no recognized chronic systemic diseases, clinical symptoms of B12 deficiency, and were not taking B12 supplementation or any drugs known to influence B12 absorption. The diet of all the participants was devoid of alcohol and none was a smoker. The diet pattern did not change during the study period, including the 15 days prior to the beginning of the study.

Baseline characteristics of the participants in each of the two intervention groups are presented in Table 1. The groups did not differ from each other in the initial biomarker values (*p* ≥ 0.1), yet a higher dispersion was observed in total plasma B12 and holoTC, if compared to the metabolites Hcy and MMA. None of the participants was clearly anemic, but mild anemia was detected in two (one male and one female) and three (one male and two female) from capsule and whey powder groups, respectively. Four subjects had baseline B12 levels above the lower reference value (150 pmol/L), but the overall cohort median (112 pmol/L, *n* = 35) was below the reference interval. Low B12 status was further supported by the overall low holoTC, as well as the elevated concentrations of MMA and Hcy (Table 1). There was no significant difference in BMI in the two groups and estimated glomerular filtration rate of all participants were above 90 (90–120) ml/min indicating normal renal function.

### 3.2. Median Biomarker Values after Four and Eight Weeks of Treatment with Capsules or Whey Powder

Table 2 presents median marker concentrations after four and eight weeks of the supplement with capsules or whey powder. The time points highlight the outcomes in the middle and at the end of treatment. Table 2 also shows a non-parametric paired comparison of baselines vs. the achieved treatment results.

The data demonstrate significant changes in all markers, though many of them still deviate from the standards accepted in Western countries, e.g., the combined indicator of B12 status ≥0.0 ± 0.5 [22]. The changes in medians for the combined vitamin score (2cB12_B12,holoTC_) and the combined metabolic score (2cB12_metabolites_) were of the same magnitude (≈+0.6) in the whey powder group. On the contrary, the treatment with CN-B12 capsules pointed to some overestimate of the vitamin score changes (≈+1.3), when compared to changes in the metabolic score (≈+0.6). A further elaboration on this subject is given in the following sections.

### 3.3. Comparison of Marker Response Curves Following Treatment with Capsules or Whey Powder

Alterations in total B12 and holoTC after supplementation of CN-B12 in capsules or HO-B12 in whey powder are depicted in Figure 2 together with their approximating curves (see Section 2.6). The parameters of best approximation are presented in Appendix A
Appendix A.

All separate mean points for B12 changes were significantly above the baseline (*p* ≤ 0.05) already after two weeks of treatment with both supplements (Figure 2a). Capsules gave a steep increase in ΔB12 (especially during the first four weeks), whereas ingestion of whey powder caused less pronounced changes. A significant difference between the two treatments was found at nearly all time points, most notably at week 2 (*p* = 0.003) and week 4 (*p* = 0.003). The overall fitting curves also revealed a significant global difference (*p* = 0.02) in the responses to the treatments with capsules as compared to whey powder.

HoloTC increased in both supplementation groups (Figure 2b). The maximal response was achieved at approximately week 4, whereupon the amplitude slightly declined. Paired comparison of week 4 vs. week 8/10 by *t*-test revealed low probabilities of equal values with *p* = 0.03/0.09 for capsules and *p* = 0.47/0.0016 for whey powder. Both the individual points and the fitting curves of the two treatments were reasonably similar. Somewhat higher levels of ΔholoTC were visible in the capsule as compared to the whey powder group at week 2 (*p* = 0.03) and week 4 (*p* = 0.02), but the two curves did not show a significant difference (*p* = 0.06).

Figure 3 and Appendix A
Appendix A present changes in the metabolites during the course of supplementation with CN-B12 capsules or HO-B12 in whey powder.

The ratio of Hcy/Hcy_0_ (Figure 3a) shows a significant reduction in Hcy concentrations (*p* < 0.05) in both groups at all time points. The curves apparently achieved a minimum at weeks four to six, but then revealed some upward drift toward baseline. Thus, probabilities of equal values at week 4 vs. week 8/10 were rather low (*p* = 0.02/0.15 for capsules and *p* = 0.013/0.0004 for whey powder). The individual points of two treatments revealed a more pronounced decrease in the whey powder group at week 2 as compared to the capsule group (*p* = 0.05). However, there was no significant global difference between the time-dependent curves for the two groups (*p* = 0.07), and the end results of the two treatments (weeks eight and ten) were identical (*p* > 0.82).

The decrease in MMA/MMA_0_ (Figure 3b) was significant for both groups (*p* < 0.01) at all time points. In our fitting procedure, we considered an apparent graphical expression of a backward trend toward baseline at weeks 4–10 (Figure 3b). However, this tendency was not clearly corroborated by paired alignment of points at week 4 vs. week 8/10 (*p* = 0.48/0.20 for capsules and *p* = 0.32/0.18 for whey powder). A comparison of the two treatments to each other at individual time points revealed only a marginal difference at week 4 (*p* = 0.05), where the decrease was slightly more pronounced for the capsule group as compared to the whey powder group. The overall curves for the two supplementations were, nevertheless, similar (*p* = 0.17), and the final results at weeks eight and ten were alike (*p* > 0.45).

### 3.4. Comparison of Response Curves for the Combined Indicators of B12-Status

The responses to supplementations were interpreted in terms of the combined indicator of B12 status (cB12) [22]. The “vitamin index” 2cB12_B12,holoTC_ included the measurements of total B12 and holoTC (Figure 4a), whereas the “metabolic index” 2cB12_metabolic_ covered Hcy and MMA (Figure 4b). Finally, changes in the combined indicator, including all four biomarkers (4cB12), are shown in Figure 4c. All variants of the combined indicator of B12 status are adjusted to the same scale [22], exposing synchronization (or its absence) between the global vitamin pool (2cB12_B12,holoTC_) and the metabolites (2cB12_metabolic_). Overlap of the two curves means that accumulation of B12 is balanced with activation of the B12-dependent enzymes. Deviation between the two curves indicates either some anomaly in the B12-processing mechanism or a short-term pre-steady state condition, when a fresh portion of B12 has not yet been converted to the cofactors (usually observed at the beginning of supplementation, e.g., at weeks 0–2).

The results presented in Figure 4 show a significant difference in the “vitamin indexes” 2cB12_B12,holoTC_ (Figure 4a), when comparing the two supplemented groups. The values of 2cB12_B12,holoTC_ were much higher after the capsule than after whey powder treatment. Irrespective of this difference, the “metabolic indexes” (2cB12_metabolic_) overlapped (Figure 4b). When superimposing the “vitamin indexes” in Figure 4a with the “metabolic indexes” in Figure 4b, one can clearly see that 2cB12_B12,holoTC_ of the capsule group considerably deviates upward from its twin capsule “metabolic index”, as well as from 2cB12_B12,holoTC,_ and 2cB12_metabolic_ for the whey powder group. In contrast, the three latter combined scores showed a good agreement with each other, indicating very similar metabolic responses for both supplementations, as well as the concurrent changes in the pools of vitamin and metabolites for the whey powder supplementation.

The 4cB12 index (that includes B12, holoTC, Hcy, and MMA) is presented in Figure 4c and shows a higher response in the capsule group than in the whey powder group. As pointed out above, this effect is driven by total plasma B12 (and to a lower extent by holoTC).

### 3.5. Forms of B12 Present on TC and HC

To further explore different responses in total plasma B12 to supplementation with capsules (CN-B12) vs. whey powder (HO-B12), we studied the distribution of the two B12-forms bound to transcobalamin and haptocorrin (Table 3). We collected and analyzed pooled plasma samples at the end of the intervention and two weeks post-supplementation. We concentrated our attention on comparison of CN-B12 and HO-B12, leaving behind cofactors (expected to be converted to HO-B12 in ambient light). The capsule group showed a high fraction of CN-B12 on both transcobalamin and haptocorrin by the end of supplementation (week 8). Two weeks later, the fraction of CN-B12 on transcobalamin decreased, while haptocorrin still harbored a relatively high amount of CN-B12. The whey powder group exhibited lower concentrations of total B12, predominantly represented by HO-B12. However, a persistent minor fraction of CN-B12 (15–20%) was also detected, assessed earlier by other authors as 14–27% [23] and 4% [24]. If we link the data in Table 3 with preceding Figure 2 and Table 2, a greater change in total B12 in the capsule group can be explained by the surplus CN-B12 circulating on haptocorrin and, to some extent, on transcobalamin.

## 4. Discussion

We studied the biomarkers, related to B12 status, in a lactovegetarian Indian population supplemented with a daily dose of 5.6 µg B12, administered as a divided portion with 10–12 h intervals (either two capsules of CN-B12 or two servings of whey powder with endogenous HO-B12). Blood samples were collected at baseline (week 0) and every second week during eight weeks of supplementation, plus two weeks post-intervention.

The present study supports and expands our previous data [8] on supplementations with CN-B12 vs. HO-B12 (the latter being either in capsules or present in cow and buffalo milk). Here, we confirm the equal value of the two vitamin forms in relation to improving metabolic biomarkers (Hcy and MMA) and underscore that measurements of just plasma B12 would give a false impression of CN-B12 superiority. For the first time, we demonstrate that the difference in plasma B12 following supplementation with CN-B12 vs. HO-B12 may well be driven by retention of unmetabolized CN-B12 on haptocorrin (and to a lesser degree on transcobalamin). Importantly, we establish that eight weeks of treatment with a high physiological dose of B12 is insufficient to normalize biomarkers of B12 status—even when dividing the dose into two daily servings in order to prevent overload of the B12 uptake system, characterized by a limited capacity for each separate B12 intake.

Our study has some limitations. The conclusions would be strengthened had the study groups been increased and the intervention period (as well as the follow-up period) prolonged. The capsule group had a rather uneven representation of men (*n* = 2) and women (*n* = 15). However, our unpublished comparison of men vs. women in previously studied cohorts [8] did not reveal any significant difference in the responses of women as compared to men. Concerning the analysis of B12 forms on transcobalamin and haptocorrin, we were not able to examine the individual samples, but only pooled samples.

A more detailed comparison of the current study to our previous works reveals that the daily dose of B12 employed in the current study (5.6 µg for eight weeks) results in a more explicit improvement in the metabolites (Hcy and MMA) than observed in previous supplementation schemes of 1.52 µg for four weeks [8] or 3 µg for eight weeks [25], Table 4. However, none of the three interventions led to complete normalization of the biomarkers, thereby suggesting that the impaired B12 status was not completely reversed by the doses administered.

The responses of metabolites and holoTC (Table 2 and Figure 2B and Figure 3A,B) showed some unexpected trends to the respective baselines at the end of our study (weeks 8 and 10). The effect was evident for Hcy and holoTC, but nearly absent for MMA (at least in statistical terms). A backward tendency of MMA was, however, noticed in our previous study [18]. We have no unequivocal explanation of these observations but can offer a plausible conjecture. Activation of a functioning enzyme indeed provides a drop in its substrate, but this drop is afterward partially reversed, if the enzyme belongs to a metabolic or/and transportation flux. This reversion originates from the fact that the decreased substrate is simultaneously the product of preceding enzymes. Lowering of product inhibition in the preceding enzyme stimulates it to produce more of the compound in question. In a well-mixed in vitro system, these down-up fluctuations are accomplished within seconds, but in a multi-compartment (multi-organ) in vivo system, the return to a new steady-state in blood might take time.

Contrary to the aforementioned markers, total B12 steadily increased and stabilized at a new steady-state level without further fluctuations (Figure 2A). The levels were, however, different for CN-B12 capsules and whey powder (containing mostly HO-B12). A more pronounced increase in total plasma B12 during supplementation of CN-B12 vs. HO-B12 (especially during the first two weeks) was consistently recorded in our current and preceding studies [8,25]. This difference in finally achieved levels depended though on the supplemented dose, showing a low difference at low doses, but a high difference at higher doses (Table 4).

Irrespective of a higher build-up in total plasma B12 upon ingestion of CN-B12, this accumulation was not reflected in the metabolic markers, which uniformly indicated that CN-B12 and HO-B12 improved B12 status with equal potency.

The aforementioned accumulation of total plasma B12 upon ingestion of CN-B12 vs. HO-B12 requires a separate discussion. First, we should point out that total plasma B12 is a sum of holoTC (present at a low concentration) and holohaptocorrin (holoHC) (present at a high concentration), reviewed in refs. [1,26,27]. HoloTC is a fast-exchanging carrier (half-life in blood ≈1 h [27]), which plays an essential role in promoting the cellular uptake of B12 through receptor-mediated endocytosis (involving a specific TC receptor found on all cells). Hitherto, no universal receptor for holoHC has been identified, and the haptocorrin-B12 complex is considered to be of limited importance for B12 tissue delivery. Haptocorrin is believed to accumulate and dispose of the inactive forms of B12, acting as a scavenging protein [27]. We speculated in our previous paper [8] that a higher total B12 in blood upon ingestion of CN-B12 might be caused by constrained intracellular processing of this vitamin form, as was the case for the nonconvertible “anti-vitamin” ethylphenyl-Cbl [28]. The cobalt site of CN-B12 is partially protected by CN-group, which can decelerate transformation of CN-B12 to the active cofactors (at least in comparison to the “unprotected” HO-B12). Such insufficient conversion apparently initiates a cyclic turnover of the unprocessed CN-B12 (cellular uptake → excretion → uptake → excretion …), leading to a gradual accumulation of CN-B12 on the slow-exchanging haptocorrin (with a half-life time in blood of approximately 10 days [27]).

Confirming our hypothesis about the accumulation of CN-B12 on haptocorrin, we here provide biochemical evidence that the high level of total B12 after administration of CN-B12 indeed originates from CN-B12 bound to haptocorrin. Over-representation of the inactive vitamin on haptocorrin implies that the total plasma B12 (holoHC + holoTC) does not reflect the real quantities of the active B12-cofactors in the tissues (if the ingested form is CN-B12). Therefore, the metabolic markers MMA and Hcy should be aligned rather with transcobalamin-bound B12 (holoTC), because this marker is apparently less sensitive to the supplemented vitamin form.

## 5. Conclusions

We report that two servings of vitamin capsules (CN-B12) or whey powder (HO-B12) for eight weeks are equally efficient in improving biomarkers of B12 deficiency in a population with low B12 status, but insufficient for normalizing B12-status. The results support the value of whey powder as a source of B12. Our observation that the capsules give a higher increase in the total plasma B12 does not signify a potential benefit of CN-B12, but simply reflects the accumulation of the unprocessed CN-B12 on the inert B12-binding protein, haptocorrin. In contrast, the changes in holoTC closely reflect the metabolic responses, irrespective of the supplemented form of B12, and can thus be recommended for monitoring B12 intervention. Further studies are needed to clarify any possible benefit of supplementation with HO-B12 as compared to CN-B12.

## Figures and Tables

**Figure 1 nutrients-11-02382-f001:**
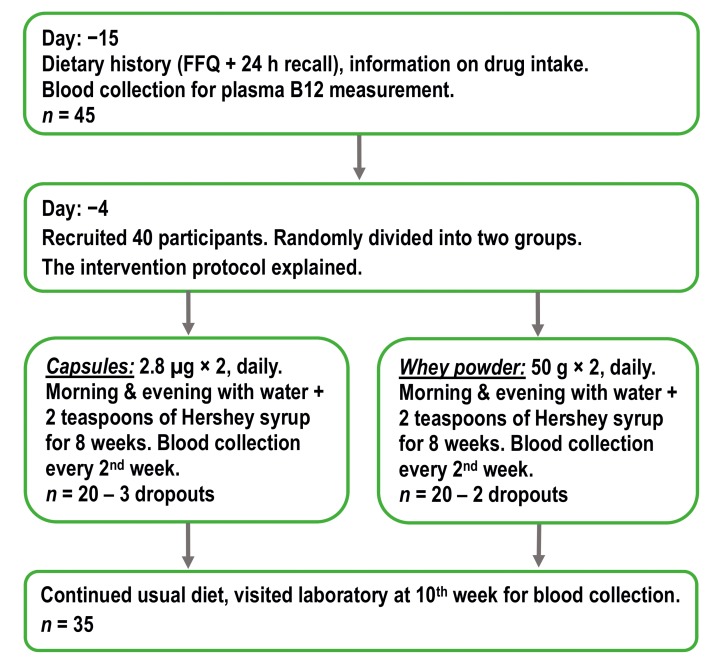
The scheme of overall study design (FFQ, food frequency questionnaire).

**Figure 2 nutrients-11-02382-f002:**
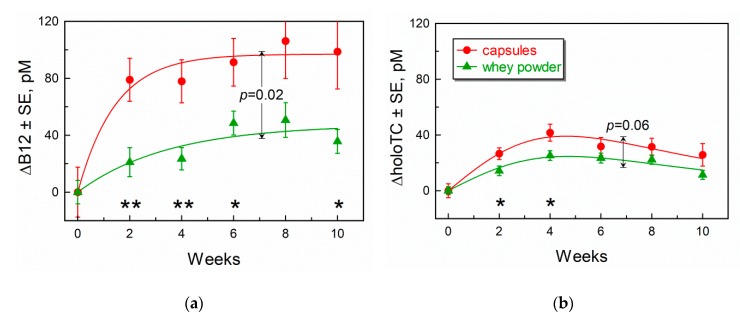
Time-dependent increase (as compared to baseline values) in (**a**) total plasma B12, fitted by Equation (1), and (**b**) holoTC, fitted by Equation (2), after supplementation of CN-B12 in capsules (●, red) or HO-B12 in whey powder (▲, green). The data are presented as mean values ± SEM. Symbols (*) and (**) indicate significant differences between the two datasets at particular time points according to *t*-test (*p* < 0.05 and 0.01, respectively). The arrows with stop-lines and accompanying *p*-values show the overall difference between the fitting curves for the two treatments.

**Figure 3 nutrients-11-02382-f003:**
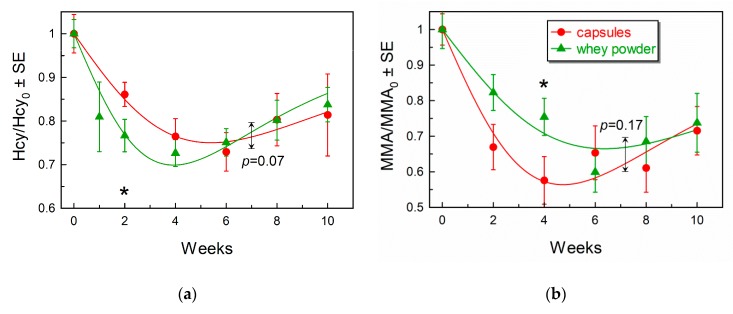
Time-dependent changes in the metabolites (**a**) Hcy and (**b**) MMA both expressed as fractions of the respective baselines (week 0) and fitted by Equation (2). Other annotations and descriptions are as in the legend to Figure 2.

**Figure 4 nutrients-11-02382-f004:**
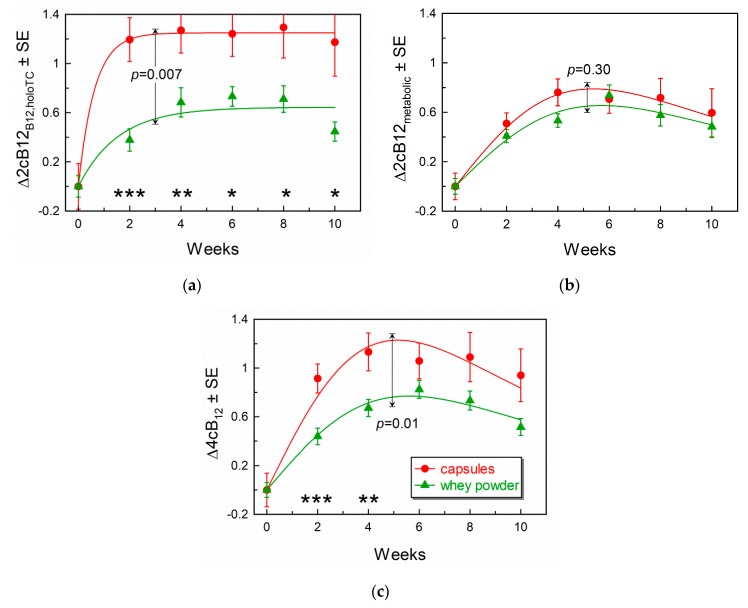
Changes in the combined indexes of B12 status during supplementations with CN-B12 in capsules or HO-B12 in whey powder. (**a**) Partial “vitamin” indexes 2cB12 for total B12 and holoTC, fitted by Equation (1). (**b**) Partial “metabolic” indexes 2cB12 for total Hcy and MMA, fitted by Equation (2). (**c**) Full indexes 4cB12 fitted by Equation (2). Other annotations as stated for Figure 2.

**Table 1 nutrients-11-02382-t001:** Concentrations of biomarkers at baseline for participants receiving capsules or whey powder supplements.

Marker	Reference Interval *	Capsules (CN-B12) Median (min/max) *n* = 17 (15 Female)	Whey Powder (HO-B12) Median (Min/Max) *n* = 18 (8 Female)	Mann-Whitney *p*-Values
P, total vitamin B12 (pmol/L)	148–630	106 (61/185)	114 (71/184)	0.10
P, creatinine µmol/L	52–110	67 (52–110)	69 (55–102)	0.56
P, holotranscobalamin (pmol/L)	35–150	19 (4/40)	23 (8/99)	0.17
P, total homocysteine (µmol/L)	5.0–15.0	19 (9/52)	17 (9/49)	0.73
P, methylmalonic acid (µmol/L)	0.1–0.28	0.81 (0.28/2.46)	0.91 (0.30/2.34)	0.69
S, folate (nmol/L)	4.54–38.6	13.5 (11.1/16.0)	13.0 (10.6/16.3)	0.78
combined index (4cB12)	−0.5–1.5	−1.87 (−3.01/−0.37)	−1.67 (−2.36/−0.33)	0.30
B, Hemoglobin (g/L) Mild anemia	M 130–180 F 120–150	128 (116/137) 1 M, 1 F	132 (122–142) 1 M, 2 F	0.67
B, red blood cell, mean corpuscular volume (fL)	80–96	82.5 (80/91.6)	81.6 (81.5/92)	0.58

P, plasma; S, serum; B, blood; m, men; w, women; *p*, probability of equal baselines (capsules vs. whey powder). * Reference intervals are those employed by the laboratory that performed the analysis.

**Table 2 nutrients-11-02382-t002:** Nonparametric comparison of the treatment outcomes at weeks four and eight vs. the respective baselines for participants receiving CN-B12 capsules or whey powder.

Plasma Markers	Capsules (CN-B12) Median			Whey POWDER (HO-B12) Median		
	0 Weeks *n* = 17	4 Weeks *n* = 17	8 Weeks *n* = 13	Baseline *n* = 18	4 Weeks *n* = 18	8 Weeks *n* = 18
Total B12 pmol/L 0 vs. 4 and 8 weeks, *p* ^1^	106	172	182	114	134	158
	–	0.00003	0.00007	–	0.004	0.001
HoloTC (pmol/L) 0 vs. 4 and 8 weeks *p* ^1^	19	59	53	23	49	48
	–	0.00003	0.00001	–	0.000008	0.000007
Total Hcy (µmol/L) 0 vs. 4 and 8 weeks *p* ^1^	18.9	13.3	15.5	16.8	13.0	13.6
	–	0.0009	0.005	–	0.0002	0.001
MMA (µmol/L) 0 vs. 4 and 8 weeks *p* ^1^	0.81	0.38	0.46	0.91	0.62	0.50
	–	0.0006	0.004	–	0.003	0.001
4cB12 0 vs. 4 and 8 weeks *p* ^1^	−1.87	−0.42	−0.63	−1.67	−0.96	−0.91
	–	0.00003	0.00006	–	0.0003	0.0005
2cB12_B12,holoTC_ 0 vs. 4 and 8 weeks *p* ^1^	−1.36	+0.08	−0.07	−0.92	−0.28	−0.24
	–	0.00003	0.00002	–	0.0003	0.00005
2cB12_metabolites_ 0 vs. 4 and 8 weeks *p* ^1^	−1.76	−0.92	−1.22	−1.94	−1.31	−1.33
	–	0.005	0.0001	–	0.000008	0.00004

^1^ probability of equal values at baseline and weeks four or eight (Wilcoxon paired test).

**Table 3 nutrients-11-02382-t003:** Distribution of CN-B12 and HO-B12 on transcobalamin (TC) and haptocorrin (HC) at the end of the intervention (week 8) and two weeks post-treatment (week 10) in the groups treated with CN-B12 in capsules or HO-B12 in whey powder.

Vitamin Form	Capsules (CN-B12)		Whey Powder (HO-B12)	
	Bound to TC ^1^, pmol/L, (%)	Bound to HC ^1,^ pmol/L, (%)	Bound to TC ^1^, pmol/L, (%)	Bound to HC ^1^, pmol/L, (%)
Week 8				
HO-B12	24 (9)	81 (30)	63 (26)	134 (55)
CN-B12	**44 (16)**	**122 (45)**	16 (7)	30 (12)
total	68 (25)	203 (75)	79 (33)	164 (67)
Week 10				
HO-B12	26 (12)	112 (52)	53 (26)	114 (57)
CN-B12	6.4 (3)	**69 (32)**	12 (6)	22 (11)
total	32 (15)	181 (84)	65 (32)	136 (68)

^1^ TC, transcobalamin; HC, haptocorrin. The analysis was performed on two pooled blood plasma samples from the capsule and whey powder groups (five participants with the maximal responses from the respective supplementation group at the indicated time points). The data represent HPLC peaks of CN-B12 or HO-B12 extracted from TC or HC and are expressed as plasma concentrations or percent (in parentheses). Percent values are related to the sum of all B12 present on TC and HC. Bold underlined values highlight the excessive content of CN-B12 (>7% on TC, >15% on HC).

**Table 4 nutrients-11-02382-t004:** Final changes in total plasma B12 and the combined metabolic index following administration of CN-B12 and HO-B12.

Dose and Time	Total Plasma ΔB12 ^2^, pmol/L, mean ± SE		ΔcB12_metabolic_ ^2^ mean ± SE	
	CN-B12	HO-B12	CN-B12	HO-B12
2 × 0.76 µg/day ^1^ four weeks [8]	+30 ± 7	+29 ± 15	+0.37 ± 0.08	+0.29 ± 0.11
	*p* = 0.95		*p* = 0.55	
3 µg/day eight weeks [25]	+55 ± 6	+37 ± 8	+0.33 ± 0.61	+0.22 ± 0.11
	*p* = 0.07		*p* = 0.86	
2 × 2.8 µg/day eight weeks (present study)	+97 ± 20	+47 ± 13	+0.70 ± 0.13	+0.59 ± 0.08
	*p* = 0.04		*p* = 0.48	

^1^ The data for HO-B12 were taken according to cow milk supplementation. ^2^ The final levels of ΔB12 and ΔcB12_metabolic_ were evaluated according to exponential approximations (Equation (1)), though the precise shapes of dependencies can be argued. Probability *p* of equal end-values for CN-B12 vs. HO-B12 data was assessed by *t*-test.

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
