# Peer review of "Cyano-B12 or Whey Powder with Endogenous Hydroxo-B12 for Supplementation in B12 Deficient Lactovegetarians"

_nutrients, 2019, doi:10.3390/nu11102382_

Round 1

Reviewer 1 Report

It has been a pleasure to read the manuscript entitled “Cyano-B12 or whey powder with endogenous hydroxo-B12 for supplementation in B12 deficient lactovegetarians”. The work presented is of high scientific value and will be of use to doctors prescribing vitamin B12 supplements to patients and to vegetarians taking supplements. The manuscript is clearly written, has an interesting study design and an innovative statistical approach to assess responses to treatment in both groups: cyano-B12 or whey powder.

The authors may wish to consider the following suggestions for improvement and clarify some points as stated below:

Page 2: change to “targeted the liver”

On page 3, Folate and B12 deficiency is defined as <4.54 nmol/L and <150 pmol/L and in Table 1 as 6.8 nmol/L and 148 pmol/L. Did the authors mean to use different cut-offs? 4.54 nmol/L is quite low for folate.

Page 3, section 2.3 Plasma EDTA is not suitable for holoTC and folate analysis on the Architect platform. Serum samples are preferable.

Page 4, second line – typo “EC6.9%?

Page 4 , line 3 and 4: consistency required “present in plasma…we prepared pools of 100 ul serum”

Table 1. Were the analyses carried out in serum or plasma?

Table 2. Specify in the legend that the numbers represent medians

Table 2 The median homocysteine increased in week 8 compared to week 4 in both groups. Can authors speculate this increase in the discussion? MMA also increased in week 8 in capsules group.

Table 3 should state that the numbers represent the mean (SD) in the legend

Author Response

Thank you for the suggestions. Suggestions and  answers have been incorporated into the text.

Page 2: Change to “targeted the liver”

The sentence is changed as suggested

Folate and vitamin B12 deficiency is defined as <4.54 nmol/L and <150 pmol/L and in Table 1 as 6.8 nmol/L and 148 pmol/L. Did the authors mean to use different cut offs. 4.54 nmol/L is quite low for folate

The values are corrected in Table 1. The cut off for serum folate in Indians is 4.54 nmol/L. The reference range is 4.54 to 38.5 nmol/L

Page 3. Section 2.3. Plasma EDTA is not suitable for holoTC and folate analysison Architect platform. Serum samples are preferable.

We used serum for folate measurements and accordingly changes have been made.  However, we used plasma samples for holoTC analysis. We used tri-level Abbott controls in every batch. Results were acceptable. Plasma was used in the previous study “Greibe et al. Increase in circulating holotranscobalamin after oral administration of cyanocobalamin or hydroxocobalamin in healthy adults with low and normal cobalamin status. Eu J Nutr. Oct 2017”. DOI 10.1007/s00394-017-1553-5

Page 4: second line: Typographic mistake; EG 6.9%

It is corrected

Page 4: consistency required” plasma and serum”

Corrected the sentence

Were the analysis carried out in serum or plasma?

Serum was used for folate and plasma was used for B12, holo-TC, homocysteine, creatinine  and MMA. Corrected in the text

Table 2: Specify that the numbers represent  the medians

Already expressed as medians in every column

The median of HC increased in Wk 8 compared to wk 4 in  in both the groups. Can authors speculate this increase in the discussion. MMA also increased in week 8 in capsule group

We add now the requested paragraph to the discussion section, where we refer to the data in Table 2, Figures 2B, 3A and 3B.

Table 3 should state that the numbers represent the mean(SD) in the legend

We regret confusion about Table 3, because the numbers are neither means nor SD. They represent evaluation of HPLC profiles obtained from the pooled blood serum samples, where the fractions of B12 bound to TC or HC were analyzed independently of each other concerning their content of CN-B12 and HO-B12. The values correspond to the HPLC peaks presented as concentrations in the original samples or in percent (in parentheses). A clarification is added to the subscript to Table 3.

Reviewer 2 Report

The current study was focused on comparison of the efficacy of whey powder containing HO-B12 with that of capsules containing CN-B12 in restoration and maintenance of B12 status in individuals following a lactovegetarian diet with a borderline B12 deficiency.The article has a number of strengths and overall has interesting findings. However, prior to publication, I would recommend several substantive changes to enhance the manuscript:

The INTRODUCTION is well written nevertheless, in my opinion, it would benefit from an overview of Vitamin B12 among individuals following a meat-free diet. The Authors should sum up some fundamental (recent) data focusing on lactovegetarian diet and B12 supplementation as well as B12 deficiency. The introduction needs to be revised to concentrate on the study aim and hypotheses (the objective and hypotheses should be formulated). Authors ought to point to the research questions.

The METHODS section should be supplemented with information regarding the recruitment and representativeness of the sample.

In the RESULTS section Authors should provide a more detailed characteristics of a sample (e.g. age, body mass index, lactovegetarian diet duration, presence/absence of another diet, presence/absence of mental disorder, etc.)

The DISCUSSION section should be more structured, according to the different research questions/hypotheses of the study and each hypothesis should include one important message, related to the findings.

Author Response

Response –Review 2

Thank you very much for reviewing and suggesting the amendments. The changes are incorporated in the text and tables

The INTRODUCTION is well written nevertheless, in my opinion, it would benefit from an overview of Vitamin B12 among individuals following a meat-free diet.

             Included in the Introduction

The Authors should sum up some fundamental (recent) data focusing on lactovegetarian diet and B12 supplementation as well as B12 deficiency.

             The daily diet of the participants included dairy products but not meat/fish. The  

             cobalamin contents of the dairy product consumed was 0.6 µg/day. Their intake was

             less than the RDA. INCLUDED in the result section

The introduction needs to be revised to concentrate on the study aim and hypotheses (the objective and hypotheses should be formulated). Authors ought to point to the research questions.

             Included in introduction

The METHODS section should be supplemented with information regarding the recruitment and representativeness of the sample.

             Only lactovegetarian Indians were recruited. Mentioned in detail in the result section

In the RESULTS section Authors should provide a more detailed characteristics of a sample (e.g. age, body mass index, lactovegetarian diet duration, presence/absence of another diet, presence/absence of mental disorder, etc.)

Daily diet of the participants included dairy products but never meat/fish. Total cobalamin content of the diet was 0.6 ug/day (lower than the RDA)

The DISCUSSION section should be more structured, according to the different research questions/hypotheses of the study and each hypothesis should include one important message, related to the findings.

            We attempt to address this shortcoming and introduce several rearrangements (and in    

            a  new section) to the Discussion section. We hope that this revision is adequate.

Round 2

Reviewer 2 Report

Thank you for making the careful revisions to this manuscript.
I accept it in present form.

Author Response

Incorporated the suggestions and answers in the text

and highlighted the changes introduced